# Food Waste Compost as a Tool of Microbiome-Assisted Agri-Culture for Sustainable Protection of Vegetable Crops Against Soil-Borne Parasites

**DOI:** 10.3390/ijms262110606

**Published:** 2025-10-31

**Authors:** Paola Leonetti, Paolo Roberto Di Palma, Giulio Gazzola, Sergio Molinari

**Affiliations:** 1Department of Biology, Agricultural and Food Sciences, Institute for Sustainable Plant Protection, Bari Unit, CNR, 70126 Bari, Italy; paola.leonetti@cnr.it; 2Waste and Secondary Raw Materials Technology Laboratory, Circular Economy Division, Sustainability Department, ENEA Casaccia, Via Anguillarese 301, 00123 Rome, Italy; paoloroberto.dipalma@enea.it (P.R.D.P.); giulio.gazzola@enea.it (G.G.)

**Keywords:** AMF, compost, marker genes, mycorrhization, PGPR, plant immune system, RKNs

## Abstract

A low-scale Food Waste Compost (FWC1), characterized by optimal physic-chemical parameters and high organic matter percentages, was used as a fertilizer and a bio-stimulant for vegetable plants. Groups of treated plants were inoculated with active juveniles of root-knot nematodes to detect the effect on plant defense. Optimal amounts of compost mixed with soil increased plant biomass 30% compared to untreated plants. Moreover, when plants were inoculated, treated roots contained about 50% less sedentary forms (SFs) of nematodes and a lower reproduction rate of the parasites than untreated plants. Although the performance of FWC1 as defense activator was similar to other microbiome-generating commercial formulations, the compost was found to be the best fertilizer in both un- and inoculated plants. Diffuse root colonization by arbuscular mycorrhizal fungi (AMF) was observed after treatments with FWC1. FWC1 water extracts did not show any toxic effect on living nematode juveniles. Expression of the marker gene of immune response *PR4b* was found to be 3–5-fold higher in the roots of inoculated plants treated with FWC1 with respect to untreated plants, thus indicating that FWC1 primes plants against RKNs (root-knot nematodes, *Meloidogyne incognita* (Kofoid White) Chitw). Data are reported to associate immunization of plants with mycorrhization occurring in FWC1-treated plants. The proposed compost is indicated as having optimal performance both as a bio-fertilizer and a bio-stimulant.

## 1. Introduction

A plant’s microbiome is its collection of plant-associated microorganism communities, also referred to as the plant’s second genome. Roots select beneficial rhizosphere microbes in naturally microbiome-rich soils; microbiome-generating formulations, containing arbuscular mycorrhizal fungi (AMF), plant growth-promoting rhizobacteria (PGPR), and other beneficial microorganisms can be added to experimental soils to test their effectiveness in stimulating plant defense against biotic stresses. The persistent use of mineral fertilizers and synthetic pesticides has increased environmental pollution and toxicological concerns, underscoring the attractiveness of these bio-stimulants and composting. Compost, along with rhizobia and other nitrogen-fixing microorganisms, organic fertilizers, and mycorrhizae, now constitute safer natural substitutes that can be used in agrosystems for healthy crop production.

Bio-stimulants are natural substances or microorganisms that are provided to plants to increase their tolerance to abiotic and biotic stresses, regardless of their nutrient content. Bio-fertilizers are a subcategory of bio-stimulants constituted by formulations of beneficial microorganisms that are able to improve the nutritional efficiency of plants and promote their growth and health [1]. Microbiome-generating composts are currently being studied as agro-products that can associate bio-fertilizing and bio-stimulating effects. Composted materials have their own microbial and fungal community and may enhance the microbial life of the soils to which they are added [2]. The addition of composted green waste (GWC) to soil may merge the benefits of these natural compounds in terms of improvement in bulk density, infiltration rate, hydraulic conductivity, water content, aggregate stability, and porosity [3]. Consequently, such improvements in the soil rhizosphere microbiome make sources of nutrients more available to plants and make plant defense responses more effective towards diseases and pests. It has recently been reported that the enrichment of soils with bacterial and fungal rhizosphere microbiomes improves the immune response in plants against soil and foliar pests [4]. More generally, plant-associated microbial communities are widely recognized to have bio-fertilizing and bio-stimulating effects on plants [5]. Therefore, the performance of a compost is linked to its maturity or curing time, which reflects the substrate composition and its microbial community [6].

In this paper, we tested the performance of a compost produced from the food wastes of a canteen (Food Waste Compost 1, FWC1) acting as a growth-promoting organic amendment and a resistance inducer against root-knot nematodes (RKNs). RKNs are the most damaging and diffused family within plant parasitic nematodes (PPNs), which include mainly root-feeder worms and represent a significant threat to global food production, with annual crop losses estimated to be 125–157 billion USD [7]. PPNs possess a hollow protrusible stylet in the oral cavity that is used to puncture cell walls, inject secretions into host cells, and ingest nutrients from cell cytoplasm. RKNs are obligatory sedentary endoparasites that enter the roots as motile second-stage juveniles (J2s) and move intercellularly through the elongation zone to reach some few cortical cells. Here, they establish their feeding sites by forming discrete giant or nurse cells that transfer solutes and nutrients from plant metabolism to the developing J2s, which soon become sedentary. Then, they develop into adult gravid females that parthenogenetically reproduce by laying 200–400 eggs in an external gelatinous matrix, which is clearly visible outside the roots as an egg mass. Moreover, the secretion of an array of effectors by nematodes induces hypertrophy and hyperplasia of the surrounding tissues, thus causing the formation of the familiar galls on the roots [8]. Damage to crops include mechanical deformation of roots, with impairment of water and nutrient uptake, and increased chance of secondary infections, which ultimately lead to plant yellowing and wilt, stunted growth, and significant productivity losses. Control of plant parasitic nematodes has generally been difficult, and the scientific community is looking for alternative low-impact management strategies such as genetic and induced resistance or the use of biocontrol agents as sustainable alternatives to toxic nematicides [9]. Symbiosis of AMF with the roots of most plants produces a mycorrhiza-induced resistance (MIR), acting against numerous different pathogens [10]. AMF belong to the phylum Glomeromycota and establish an intimate relationship by entering root cells and developing specialized structures in the cytosol known as arbuscules [11]. AMF-containing commercial formulations have been reported to be effective in limiting nematode infection through root AMF colonization, which activates the plant immune system, triggering a prompt and valid defense response upon the nematodes attempt to establish themselves [4,12,13]. However, such formulations did not show a satisfying plant growth-promoting effect, especially for plants that were not challenged by nematodes.

It has long been known that organic soil amendments stimulate the activities of microorganisms that are antagonistic to PPNs. The process of composting waste from industrial, agricultural, biological, and other activities with the decomposition of organic matter results in the accumulation of specific compounds in the soils that may be toxic to nematodes. Moreover, the addition of compost has been reported to induce improvements in soil structure and fertility, resistance to diseases and pests, release of parasites (fungi and bacteria), and other nematode antagonists [14]. More recently, the attempts of using composted municipal green waste to control PPNs have been reviewed [2]. Recently, two bacterial communities obtained from a compost provided to tomato markedly inhibited reproduction and root invasion by *Meloidogyne javanica* [15].

Starting from the successful results obtained by treating vegetable plants with AMF-containing commercial formulations in terms of nematode control, in this paper, FWC1 was compared with such formulations as a bio-stimulant, tested alone or mixed with minimal amounts of the formulations. FWC1 was found to be a bio-stimulant that is as good as the other tested commercial products, and it additionally showed a consistent bio-fertilizing effect both on healthy and nematode inoculated plants. Finally, the mechanisms through which the compost induces the immunity of vegetable plants to RKNs were found to be triggered by AMF colonization of roots with augmented expression of the genes markers of a hypersensitive response during the early stages of nematode infection.

## 2. Results

### 2.1. FWC1 as a Bio-Fertilizer and Bio-Activator of Plant Defense Against RKNs

FWC1 was first tested on healthy tomato (*Solanum lycopersicum* L.) plants as a bio-fertilizer and on plants inoculated by RKNs as a bio-stimulant of plant defense. Plant growth factors were detected 50 days after treatments, whilst plant growth and nematode infection factors were detected 50 days after nematode inoculation; inoculation was carried out 7–10 days after the treatments. The effects of FWC1 on growth and infection were determined according to plant age. Seedlings (average weight at treatment: 1.5–3.0 g) and juvenile plants (average weight at treatment: 5.0–7.0 g) were planted in soils enriched with 10 and 30 g kg^−1^ FWC1, respectively. FWC1 treatment markedly promoted the growth of healthy seedlings, whilst SH and SW of healthy juvenile plants showed a much lower increase when pre-treated with FWC1 (Table 1). Therefore, juvenile plants were pre-treated with a mix of FWC1 and one AMF-containing formulation (Ozor) at a low dose to augment the positive impact of FWC1 on shoot growth. This addition increased shoot growth with respect to untreated juvenile plants, although not at the level observed in seedlings treated with FWC1 only. Moreover, the addition of Ozor had a negative impact on root development. FWC1 had a weaker bio-fertilizer effect on nematode-infected tomato plants. It caused only about 30% SW increase in nematode infected seedlings. No support of growth was observed in juvenile plants; however, when FWC1 was mixed with Ozor, both SW and RW showed a 35% increase compared with untreated plants.

FWC1 induced a consistent decrease in both reproduction rate and plant damage in seedlings inoculated with nematodes. Again, the bio-stimulating effect observed in juvenile plants was not as marked as that observed in seedlings (Table 1). In this case, the addition of Ozor to FWC1 did not enhance the bio-stimulating effect caused by FWC1. In general, reduction of EMs and SFs caused much lower competition for food of the successful females, which almost doubled the amount of eggs laid (female fecundity, FF).

The effective doses of FWC1 were fixed according to the age and size of the plants at treatment as 10 and 30 g kg^−1^ soil FWC1 for seedlings and juvenile plants, respectively. Treatments at lower doses, such as 5 and 10 g kg^−1^ soil FWC1, did not induce any effects on juvenile plants in terms of plant growth and infection, and they had slightly positive effects on seedlings (Appendix A). On the other hand, FWC1 in excess (15 g kg^−1^ soil) had a negative effect on the growth of infected tomato seedlings (Appendix A). Previous tests for the most effective doses of FWC1 to be used on each crop–nematode system, considering the size of the plants to be treated, are therefore mandatory.

Since seedlings of tomato had been found to be more sensitive to FWC1 than grown-up ones when inoculated with nematodes, seedlings of pepper (*Capsicum annuum* L.) treated with suitable amounts of FWC1 (6 g/kg soil) were also tested. Additionally, FWC1 was mixed with minimal amounts (one tenth of the most effective dosage) of the commercial products Myco and Flortis to search for putative cumulative effects on nematode infection and plant growth (Table 2). Again, FWC1 treatment sustained plant growth and reduced nematode infection rate almost as much as it did for tomato seedlings. The addition of low amounts of Myco and Flortis to FWC1 did not substantially improve the performance of the sole compost.

When FWC1 was provided in excess to pepper seedlings (10 g kg^−1^ soil), the plant growth-promoting effect was annulled, and nematode infection was reduced at a much lower rate (Appendix A).

The effectiveness of FWC1 to act as a defense bio-activator against RKNs on pepper seedlings was compared with three other microbiome-generating commercial formulations, Ozor, Myco and Flortis, at doses that had been found to be effective in reducing nematode infection (Table 3). FWC1 treatment showed a consistent growth-promoting effect that was not present when the other formulations were provided; conversely, its ability to control nematode infection was similar.

Afterwards, the durations of the effects of single pre-treatments of FWC1 were tested by cropping egg plants followed by tomato plants, without any additional second treatment. Tomato was planted in the same untreated and treated soil where egg plants had been planted and inoculated about 2 months earlier; tomato plants were re-inoculated a few days after planting. The test was performed to find out the duration of FWC1 in its active form in the soil (Table 4). Eggplant (*Solanum melongena* L.) was the most sensitive crop to FWC1 as a bio-stimulant of plant immunity against RKNs; in this crop, FWC1 more than halved the severity of nematode infection compared with that found in untreated plants. However, the effect of the single pre-treatment made on the eggplants markedly diminished for tomato, the second crop, thus indicating that a single pre-treatment is suitable for one crop and should be repeated on the following crop to give optimal results.

### 2.2. Mechanisms Through Which FWC1 Acts as a Bio-Activator of Plant Defense Against RKNs

One month after FWC1 treatment, tomato roots were found to be diffusedly colonized by AMF (Figure 1). When the number of AMF-infected areas per gram of root fresh weight was counted under a stereoscope in plants treated with FWC1 or Ozor, the levels of AMF colonization were found to be very similar and both largely exceeded the level found in the control plants (Figure 2).

Other tests were arranged to determine if FWC1 had a direct toxic effect on nematode invading forms (J2s). Aqueous extracts of FWC1 were prepared and diluted 1:10 and 1:100, in which J2s were incubated for 48 h (Figure 3). Survival rates of J2s were measured after 24 and 48 h of incubation and compared with those of J2s incubated in water. No significant differences were found, thus indicating the absence of toxic effects of the formulation against J2s.

### 2.3. FWC1 Induces Plant Immune Response Against RKNs

The expression of key genes, markers of induction of plant immune response against RKNs, was detected using qPCR (quantitative Real-Time PCR) in tomato roots 7, 14, and 21 days after nematode inoculation. Plants treated with FWC1 and a mix of FWC1 + Ozor were compared with untreated control plants. One of the tested genes was *PR4b*; its over-expression is indicative of hypersensitive cell death induction, and it is generally a marker of the execution of plant immunity following a biotic challenge [16]. In the successful RKN–tomato interaction occurring in control plants, the expression of the gene was markedly repressed until 14 dpi; conversely, when plants were treated with FWC1 and FWC1/Ozor, this gene was generally over-expressed, thus indicating the induction of plant immunity in treated plants (Figure 4a). For successful development in roots, nematodes induce the expression of genes encoding for antioxidant enzymes that actively detoxify reactive oxygen species, and in particular, hydrogen peroxide (H_2_O_2_). We tested two genes encoding the antioxidant enzymes glutathione peroxidase (GPX) and catalase (CAT), particularly active in H_2_O_2_ degradation [17]. As predicted, in the early stages of nematode attack, expressions of GPX and CAT were highly enhanced in the controls (Figure 4b,c); however, when plants were pre-treated with FWC1 and FWC1/Ozor, such enzyme-encoding genes were significantly down-regulated, thus indicating that the interaction with these compounds induces a peroxidative environment in roots at the early stages of nematode attack, which is detrimental to the pest’s development and can be considered as a process occurring in the plant immune response.

## 3. Discussion

It is generally known that a plant growth-promoting effect is exerted by beneficial soil microorganisms, among which AMF represents a key functional group supporting plant growth, nutrition, and health [18]. Different AMF species and isolates may have different colonization ability and efficiency; moreover, the colonization ability and efficiency of microbiome-generating formulations may be mediated by the diverse and abundant bacterial communities associated with mycorrhizal roots, spores, and extra-radical hyphae. Commercial AMF formulations have recently been reported to contain bacteria isolates showing the best combination of PGP (plant growth promoting) traits [19]. The food waste compost studied in this paper has been shown to contain AMF that readily colonize roots at levels similar to a commercial AMF formulation (Ozor); the ability of another commercial AMF formulation used in this study (Myco) to actively colonize tomato roots has already been reported [13]. Although the colonization level appears to be the same between the compost and other microbiome-generating formulations, in the absence of nematode challenge, FWC1 was the only formulation to have a consistent PGP effect on horti-crops [4]. Therefore, FWC1 could also be used as a bio-fertilizer and an organic amendment, since its composition suggests it may enrich soils with both beneficial minerals, such as phosphorus (P), nitrogen (N), potassium (K), calcium (Ca), copper (Cu), zinc (Zn) and iron (Fe), and AMF to facilitate the uptake and transfer of these mineral nutrients from the soil to the plants by means of the extraradical mycelium (ERM) extending from colonized roots. Moreover, the content in humic (HA) and fulvic acids (FA) found based on the physicochemical analysis (16.3%), and consequently its humification degree (DH, 82%), is much higher than that commonly reported for commercial organic and composted fertilizers. FWC1 shows a high degree of resistance to decomposition, indicating good maturity and stability. It can be stored at 8–10 °C for at least 2 years, maintaining its effectiveness (authors communication). Maturity and stability are important factors that qualify one compost as effective to be used as soil amendment, organic fertilizer, and a nutrient source for plant growth and health [20]. According to these many optimal factors, FWC1 can be indicated as a product that consistently enhances the growth of healthy and nematode-infected plants. The careful and well-planned addition of composts with the characteristics of FWC1 to soil is likely to be an important tool to improve the environmental aspects of agriculture as well as to promote the production of healthy crops for food safety. Moreover, it has been evidenced that this compost may be made even more effective as a bio-fertilizer by adding minimal amounts of suitable AMF-containing formulations, as occurred with Ozor and shown herein.

The data of this study suggest that the fungal/bacterial strains associated with FWC1 may have a better combination of PGP traits than the other tested commercial formulations. Therefore, further studies are planned to characterize and possibly isolate the main AMF and bacterial strains contained in FWC1 and in the rhizosphere of plants treated with FWC1. Modulating the quantity and quality of the microorganisms involved in the composting process will be useful in the search for an enhancement in the already good performance provided by FWC1. 

This line of research is within the framework of microbiome-assisted agriculture with the aim to produce marketable natural formulations to be used as both biofertilizers and bio-stimulants for supporting sustainable crop production [5].

In addition to being a good biofertilizer, FWC1 was found to be a defense activator and to markedly reduce nematode infection. Generally, addition of composts to agricultural soils has been reported to contribute to suppression of soil-borne plant pathogens and pests, plant parasitic nematodes included. This suppression is due to the additional nitrogen and its mediated release of allelochemicals generated during product storage or by subsequent microbial decomposition [21]. A low C:N ratio (<20), such as that characterizing our FWC1 (13), has been found to be the most suitable in soil amendments to have a suppressive effect on nematode infection [22]. The proposed underlying mechanisms of such an effect can be different, such as systemic induction of the plant immune system, stimulation of nematode natural enemies, and release of nemato-toxic compounds such as ammonia and urea [23,24]. The data presented herein strongly indicate that FWC1 contains AMF that colonize roots, and this colonization may trigger the *priming* of plants that results in a prompt immune reaction in the early stages of nematode attack. It has long been known that mycorrhiza-induced resistance (MIR) can protect against root nematodes and miner and generalist chewing insects in above-ground tissues [10]. The molecular mechanisms associated with MIR have previously been revealed using the AMF-containing formulation, named herein as Myco, provided to tomato plants before nematode inoculation [12,13]. Myco-mediated MIR against RKNs was found, in roots at the earliest stages (3, 7 dpi) of nematode attack, to be associated with overexpression of the gene *PR-4b*, which encodes for the cell death-inducing pathogenesis-related protein PR-4b, and with a down-loading of *GPX* and *CAT*, the genes encoding for the most active anti-oxidant enzymes glutathione peroxidase and catalase, respectively. Conversely, in a normal compatible plant–nematode interaction, *PR-4b* expression was highly repressed and *GPX/CAT* expression was highly induced. In this study, the expression of the same above-mentioned genes was detected during a longer time course after inoculation (7, 14, 21 dpi) in the roots of tomato plants treated with FWC1 and a mixture of FWC1 with small amounts of Ozor. At 7–14 dpi, FWC1- and FWC1/Ozor-mediated plant reaction to RKNs caused gene expression changes very similar to those produced by Myco-mediated MIR; greater changes in gene expression were not detected in the roots of control or treated plants at 21 dpt. Then, it can be asserted that MIR has an important role in the mechanisms underlying resistance induction against RKNs in plants provided with both AMF-containing commercial formulations and FWC1. On the other hand, the addition of Ozor to FWC1 did not enhance defense gene expression, and it did not increase the reduction of the biological infection parameters caused by the compost; the addition of Ozor markedly increased only the fertilizing effect in tomato juvenile plants. Conversely, the addition of small amounts of Myco and Flortis did not result in a substantial improvement in the FWC1 performance both as a biofertilizer and as a defense inducer.

Nematode infection, then, is reduced because of an immune response triggered by plants primed by the presence of compost in the soil. A direct toxic effect of FWC1 on J2s seems unlikely under our experimental conditions, although previous studies reported nematocidal activities of compost water extracts against RKNs [19,25]. Conversely, in our experiments, J2s incubated in FWC1 water extracts apparently did not cause any change in J2 mortality up to 48 h after incubation with respect to J2s incubated in water. Therefore, although FWC1 may be not toxic to nematodes, an impairment of juvenile movements in soils of FWC1-treated plants cannot be ruled out.

The effect of Myco and Ozor had already been tested on tomato plant growth and defense response efficiency in nematode-inoculated plants [4]. The fitness costs of resistance induction, due to the diversion of energy metabolism from growth and development to defense, were evident, in that the consistent relief of symptoms did not result in an enhancement of fitness in the infected plants. In this study, Myco and Ozor were again tested in pepper seedlings, along with Flortis with constituents similar to those of Myco, and compared with FWC1. FWC1 was found to be approximately as effective in infection reduction as the other formulations. Conversely, treatments with FWC1 induced a remarkable increase in both shoot and root weights compared with control plants. In this case, the possible fitness costs of resistance induction were largely overcome by the PGP effect of the compost. Therefore, FWC1 acts both as an organic amendment supporting plant development and, through its microbial components, as a MIR-inducer, just as the other commercial formulations. The benefit given to plants as a growth-promoting agent exceeds the fitness costs due to the defense strengthening.

The amount of compost mixed into soil in terms of g kg^−1^ is extremely important. Low doses or doses in excess can impair the benefits brought by the addition of compost to soils, exactly as has been reported for other microbiome-generating commercial formulations [13]. The conditions of treatments with FWC1 should be carefully evaluated according to the specific crop–pest system, and more importantly, to the size and age of the plants to be treated. Young plants respond better to the treatments if suitable doses are provided. The same doses provided to juvenile plants do not work. Higher doses that are effective for juvenile plants can be phytotoxic to seedlings. Therefore, screening of the best doses should be carried out before the application of these microbiome-assisted tools in agronomic practices.

It has already been suggested that FWC1 acts as a plant defense activator through the AMF root colonization mediated by its associated microbiome. This ability through AMF is likely to depend on their colonization capacity, that is, the amount of root colonized within a certain time, or the time taken to colonize part of roots by a determined dose of inoculum [26]. Therefore, it is likely that the AMF contained in FWC1 must be provided to plants in suitable amounts to give the highest colonization capacity, that is, the fastest and most diffused colonization process relative to different root extension, consistence, exudates composition, etc., which vary with plant age. Consequently, the effectiveness of FWC1 as an immunity primer against RKNs depends on the quality and quantity of the contained AMF, as well as on the time allowed for fungi to establish symbiosis with roots before a nematode attack. In this study, under our controlled environmental conditions in a glasshouse, FWC1 treatment 7–10 days before nematode inoculation seemed to be a suitable time interval to allow for root colonization by AMF and the *priming* effect, provided that the optimal dose had been used. Conditions that mimic fields in which nematodes are present before planting and treatments should be tested in future trials. However, it is well known that treatments with immune system activators must be preventive of pest attack; usually, curative interventions, when infection is settled, are not effective. Therefore, compost spread in fields should always precedes planting. Once mixed with soil at the most suitable time and in appropriate amounts, FWC1 works well with the first planted crop; if no additional treatment is provided, its effectiveness decreases over time and is annulled with the second crop planted months later. Evidently, the amount remaining alive and effective in soil after the first harvest decreases to a level that is not sufficient to provide the best results.

Further studies are required to establish the exact conditions to use food waste composts as key elements of integrated pest management within the frame of microbiome-assisted agriculture. This study shows that negative reports may result from trials performed without taking into account these exact conditions.

## 4. Materials and Methods

### 4.1. Production and Chemo-Physical Characteristics of FWC1

The compost, named in the text as Food Waste Compost 1 (FWC1), used in the experimental tests was produced at the Casaccia Research Centre (CRC) of the Department for Sustainability of ENEA, Rome, Italy. The equipment used were two electromechanical composters (EC, model Big Hanna T60 (https://www.bighanna.com/e_prod/, accessed on 27 October 2025) for the autonomous management of biowaste produced by the canteen of the CRC. The schematic diagram of the EC is shown on the above-mentioned website where the structure of EC is described: it consists of a single 1 m^3^ rotating cylindrical chamber with no internal mechanical components in motion, and it is continuously fed. The rotating chamber and the new feed push the organic material towards the outlet at the opposite side of the inlet. A fan ensures air ventilation through the organic matter inside the composter, and the exhaust air is sent to a biofilter for odor removal. The active compost coming out of the EC is moved to a heap, where it undergoes a curing phase. The input capacity of the EC is 25–30 kg per day, and the filling grade is 60–70%, which corresponds to a storage capacity of 350–450 kg [27,28]. For this work, the EC was loaded daily (from Monday to Friday) with 15–20 kg of biowaste and 2.2–5.5 kg of bulking agent (pruning of Arundo donax canes, shredded into pieces of 2–5 cm, which represented 15–20% wt of the food waste) for 60 days and completely emptied on the 91st day to ensure an average residence time for all of the organic matter. At the beginning of the experiment, 25 kg of mature compost was uniformly distributed inside the EC chamber as a microorganism inoculum. The biowaste used consisted of leftovers and kitchen scraps from the research center canteen. The FWC1 curing phase consisted of periodical overturn and watering of the heap for an additional 120 days. Finally, the mature compost was sieved using a Scheppach RS350 automatic rotary sieve fitted with a 10 mm mesh. The sieved compost was used for experiments and for physico-chemical characterization.

Water content was determined by drying at a constant temperature of 105 °C until a constant weight was reached, measured with a thermo-balance (Crystal Therm, Gibertini, Novate Milanese, Italy), while pH was monitored using a portable meter (HI99121, Hanna Instruments, Singapore) specifically designed for soil analysis. Total carbon and total nitrogen content was determined using an elemental analyzer (Elementar, Langenselbold, Germany, vario MACRO), while heavy metals (Cd, Cr(VI), Cu, Fe, Mn, Ni, Pb, and Zn) and microelements (Ca, P, K, and Mg) were quantified with inductively coupled plasma (ICP-OES, Perkin Elmer-Optima 200DV, PerkinElmer, Singapore) after carrying out the total dissolution of each sample using a micro-wave-assisted acid digestion procedure. The content of mercury (Hg) in the samples was measured directly using an AMA—254 (FKV, Milestone, Toronto, M5H 4E3, Canada) spectrometer. All the results are expressed as % of dried weight.

The analysis of the total extractable carbon (TEC) and the humified fraction, consisting of HA and FA, was obtained via extraction in a solution of 0.1M NaOH and Na_4_P_2_O_7_ × 10 H_2_O. After centrifugation (3000 g for 2 min) and filtration of the supernatant through 0.8 mm cellulose acetate filters, TEC was determined for the obtained extract via oxidative digestion with 50% K_2_Cr_2_O_7_ in H_2_SO_4_ and titration of the residual dichromate with Fe(NH_4_)_2_(SO_4_)_2_ × 6 H_2_O. The humified fraction was obtained from one portion of the extract by recovering the HA via acidic precipitation, then separating the FA from the non-humified fraction (NH) using chromatography with a column filled with PVPP (polyvinylpolypyrrolidone). The adsorbed FA was eluted from the PVPP column using a 0.5 M NaOH solution and was added to the previously precipitated HA to determine the humified fraction content (HA  +  FA) using the same method used for TEC detection. Finally, the degree of humification (DH) was reported as follows:$$\%DH=\left[\frac{HA+FA}{TEC}\right]\times 100$$

The composition of FWC1 is reported in Table 5.

### 4.2. Treatment of Plants with FWC1, AMF, and Mixed Formulations of FWC1 and AMF

Seeds of the tomato (*Solanum lycopersicon* L.) cultivars Roma VF, Regina, Fiaschetto, and Marmande, of the eggplant cv. Black Beauty, and of the pepper cv. Quadrato d’Asti, all fully susceptible to root-knot nematodes (RKNs), were surface-sterilized and sown in a sterilized mixture of peat and soil. Seedlings were transplanted into plastic boxes (cm 50 × 30 × 12 H) or 110-cm^3^ clay pots filled with a freshly field-collected loamy soil and located in temperature-controlled benches (soil temperature 23–25 °C) in a glasshouse; 11 plants were planted in each box, with 1 plant per pot. To detect the best dosage for inducing a resistance effect against the tested pests, increasing amounts of FWC1 (1–100 g kg^−1^ soil) were mixed into the soil before planting. Plants were divided into seedlings and juvenile plants with a weight range at treatment of 1.5–3.0 and 5.0–7.0 g, respectively.

For comparison, 3 microbiome-generating commercial formulations were used:-Ozor (Bioplanet, Cesena, Italy), containing 500 propagules g^−1^ of *Glomus intraradices* CMCCROC7;-Micosat F^®^ (named Myco in the text, C.C.S., Aosta, Italy);-Flortis Bio (named Flortis in the text, Orvital S.p.A., Milan, Italy); the last 2 formulations contained 40% of mixed *Glomus* spp.

The FWC1 doses used in the treatments were 6.0 and 10.0 g kg^−1^ soil for the pepper and tomato seedlings, respectively. For the eggplants, the effective dosages were those used for the tomato seedlings. FWC1 was tested on pepper seedlings alone or mixed with minimal amounts of Myco and Flortis (both 0.5 g kg^−1^ soil). Pepper seedlings were also treated with the most effective dosage ranges of Ozor, Myco, and Flortis (0.3–0.5, 5.0–6.0, 5.0–6.0 g kg^−1^ soil, respectively). Juvenile tomato plants were provided with 30 g kg^−1^ FWC1 or with FWC1 mixed with Ozor 0.6 g kg^−1^, a dose lower than the most effective one for juvenile plants (1.5 g kg^−1^). Treatments with microbiome-generating formulations were carried out 7–10 days before nematode inoculation to allow for the establishment of beneficial interactions between mycorrhizal microorganisms and roots. For all tests, FWC1 was previously filtered through 500 mm filters to discard coarse components before treating the plants.

### 4.3. Root AMF Colonization

Tomato plants treated with FWC1 or Ozor at their most effective dosages, and others left untreated as a control, were grown for 40 days after treatment (dpt). Then, the roots were collected and stained to detect AMF colonization using the lactophenol blue method [29]. Soil particles and debris were gently removed from freshly washed roots, which were chopped into short pieces. Pieces coming from each treatment were divided into 3 small beakers, considered as 3 replicates. Root samples were immersed in 10% KOH, and beakers were placed into a water bath located under a fume hood for 45 min at 90 °C. The roots were thoroughly rinsed with water and acidified in 1% HCl. Staining was performed by soaking the roots in 0.05% lactophenol blue in a 90 °C water bath for 30 min and destained in a mix of 30% methanol and 10% acetic acid at room temperature. Destained roots could be stored in a refrigerator until being observed under a dissecting microscope (Leica M 125, Leica Microsystems, Wetzlar, Germany) with a LEICA IC80 HD photocamera (Leica Microsystems, Wetzlar, Germany). Discrete blue-stained areas containing typical fungal structure (i.e., hyphae, vescicles/spores, and arbuscules) were observed and counted per root fresh weight unit to measure the success of root colonization.

### 4.4. Nematode Inoculation and Determination of Infection Level

Active second-stage juveniles (J2s) of RKNs (*Meloidogyne incognita* (Kofi et White) Chitw. were constantly available for experiments on the inoculation of compost-treated and untreated plants. Egg masses containing hatching eggs were recovered from the infested roots of susceptible vegetable plants reared in a glasshouse and incubated on 500 mesh sieves in tap water at 25 °C in the dark. After 2–3 days, J2s were collected, and after dilution, amounts of J2s mL^−1^ were measured under a dissecting microscope at 25× magnification. Plants were inoculated with 500–1000 J2s each by pouring suitable volumes of a stirring J2 suspension into 2 holes made in the soil at the base of the plants. Inoculations were carried out 7–10 days after the treatments.

Under the adopted experimental conditions, inoculated J2s turned into sedentary forms (SFs: J3s, J4s, swollen females), which developed into gravid females that laid eggs 30–40 days after inoculation. The newly hatched second generation J2s in the soil were then able to re-infest roots and turn into SFs. However, plants were harvested 40–50 days after inoculation, before this second generation was able to reproduce. Most of the individuals of this second generation were counted as SFs. The level of severity of nematode infection was measured according to nematode reproduction rate and level of damage to the roots in terms of root galling. The level of root galling was considered to be proportional to the numbers of SFs and expressed as SFs g^−1^ of root fresh weight (rfw). SF number is a more affordable and statistically evaluable indicator of galling state and damage level of the roots. In most of the bioassays, reproduction rate was expressed as the numbers of egg masses (EMs) g^−1^ rfw, reproduction potential (RP), and female fecundity (FF). The last 2 indices were calculated as follows:RP = *P_f_/P_i_*; FF = eggs/EMs
where *P_f_* (final population) is the number of eggs per root system, and *P_i_* (initial population) is the number of inoculated J2s.

Plant growth changes due to compost/AMF treatments were detected by measuring shoot (SW) and root (RW) weights plus shoot height (SH) at harvest. Nematode life stages were extracted from the roots excised from the harvested plants, washed free of soil debris, and chopped into fragments. Each sample for extraction was arranged using 2 root systems. Three sub-samples were separated and weighed for the extraction of EMs, SFs, and eggs. EMs were detected by immersing the roots in 0.1 g L^−1^ Eosin Yellow in a refrigerator for at least 1 h. After incubation, gelatinous masses were easily observable as red colored under a stereoscope (6× magnification) and manually counted and isolated. SFs were extracted by incubating roots in a mixture of the enzymes pectinase and cellulase at 37 °C in an orbital shaker to loosen the bindings between sedentary nematodes and roots. Afterwards, roots were ground in physiological solution, and sedentary forms were collected on a 250 µm sieve. Aliquots (2 mL) of stirring SF suspensions were pipetted in small Petri dishes, and their numbers were counted under a stereoscope (12× magnification). Eggs were extracted by stirring root samples in diluted bleach and were counted under a stereoscope (25× magnification) [30].

### 4.5. Tests of FWC1 Toxicity to Nematodes

For toxicity tests, FWC1 water extracts had to be prepared. FWC1 powder was suspended in distilled water (0.3 g mL^−1^). The suspension was incubated for 3 days in an orbital shaker at 25° C. The suspension was first filtered using gauze, then centrifuged for 10 min at 500 g, and lastly filtered through filters of size-decreasing pores. Final filtration was carried out using 0.45 µm nitrocellulose filters. FWC1 water extracts (0.3 g mL^−1^) were diluted 1:10 and 1:100 and used to test FWC1 toxicity to nematodes. Freshly hatched J2s in tap water were used in assays of J2 survival in FWC1 diluted water extracts. J2s were concentrated by filtering through 5.0 µm cellulose nitrate membrane filters. A similar number of J2s (approx. 11,000–13,000) was suspended in flasks filled with 200 mL of distilled water and 1:10 or 1:100 diluted FWC1 extract. Flasks were placed in an orbital shaker in the dark at 25° C. One-mL aliquots of stirring suspensions from FWC1-treated J2s and controls were loaded onto special glass slides after 24 and 48 h incubation. J2s were analyzed under a stereoscope (25× magnification). Those individuals that did not move and had a rod-like shape were scored as dead; surviving individuals had clear typical movements. Data are expressed as percentages of survivors with respect to the numbers of living juveniles detected before the incubation.

### 4.6. RNA Extraction, cDNA Synthesis, and Quantitative Real-Time Polymerase Chain Reaction

Roots from plants treated with FWC1 and FWC1 + Ozor, as well as untreated plants, were collected at 7, 14, and 21 days after inoculation (dpi). Two biological assays were done; in each biological assay, samples were collected from the roots of 6 plants per treatment—3 RNA extractions were perfomed on each root sample; six replicates were obtained and used to have the means ± SD per treatment. Root samples were weighed and immediately used for RNA extraction or stored at −80 °C. First, roots were placed in a frozen porcelain mortar into which liquid nitrogen was poured. Aliquots of ground tissue (100 mg) were used for RNA extraction. Extractions of total RNA were carried out using an RNA-easy Plant Mini Kit (Qiagen, Hilden, Germany) according to the instructions specified by the manufacturer. RNA quality was verified based on electrophoresis runs on 1.0% agarose gel and quantified using a Nano-drop spectrophotometer. cDNA synthesis was carried out from 1 μg of total RNA using a QuantiTect Reverse Transcripton Kit (Qiagen, Germany) with random hexamers according to the manufacturer’s instructions. PCR mixtures (20 μL final volume) contained RNAse-free water, 0.2 μM each of forward and reverse primers, 1.5 μL of cDNA template, and 10 μL of SYBR^®^ Select Master Mix (Applied Biosystems, Buccinasco, Italy). PCR cycling consisted of an initial denaturation step at 95 °C (10 min); 40 cycles at 95 °C (30 s), at 58 °C (30 s), and at 72 °C (30 s), with a final extension step at 60 °C (1 min). qRT-PCRs were performed in triplicate using an Applied Biosystems^®^ StepOne™ instrument (Applied Biosystems, Singapore). The following tomato genes were tested: glutathione peroxidase (XM_004244468.3, GPX), pathogenesis-related gene 4b (NM_001247154.1, PR-4b), and catalase 2 (NM_001247257.2, CAT2). For each oligonucleotide set, a no-template water control was used. Actin-7 (NM_001308447.1, ACT-7) was used as the reference gene for quantification, as it was found to be the most suitable one for the experimental conditions used in this work. The oligonucleotide primers for each gene are described in Appendix A.

The threshold cycle numbers (Ct) for each transcript quantification were examined, and the relative fold changes in gene expression between uninfected and uninfected roots, treated or not, were calculated using the 2^−∆∆CT^ method [31].

### 4.7. Experimental Design and Statistical Analysis 

Three different bioassays were performed to test FWC1, Ozor, Myco, and Flortis as activators of defense against nematode infection and as biofertilizers; one bioassay consisted of 6 treated and 6 untreated inoculated plants, used as controls. Means of plant growth parameters are the results of 18 replicates, coming from 3 experiments. Conversely, as it concerned infection level, one value of each infection factor came from 2 root systems; therefore, means were obtained from 9 replicates. Means ± standard deviations of control and treated plants were separated based on a paired *t*-test (** p* < 0.05) when just one treatment was compared with one control and based on a Duncan test (** p* < 0.05) when more than one treatment was compared with one control using MS Excel software. The numbers of AMF-colonized root areas were expressed based on a fresh weight unit base. Values are the means of 6 different measurements at the stereoscope ± standard deviation; experiments to test AMF colonization of roots were designed to use 12 plants for each of the 3 tested treatments. Means ± standard deviations were calculated from 6 colored root samples; the means of control plants were separated from those of treated plants based on a Duncan test (** p* < 0.05) using MS Excel Software. The values of the toxicity tests, in terms of % J2 survival, were recovered from triple counts at the stereoscope; means ± standard deviations were obtained from 3 values, and the significance of the difference between the means of the control surviving J2s and those of the J2s incubated in the FWC1 extracts was verified based on a paired *t*-test. For RNA extraction, plants coming from 2 independent bioassays were used; roots from 2 plants of the same treatment constituted one sample; RNA was extracted from 3 different samples of roots per treatment, harvested at each dpi. qRT-PCR data are expressed as means (*n* = 6) ± standard deviations of 2^−ΔΔCt^ values of each group from treated/inoculated plants, considering as 1 the values of each group from untreated/inoculated plants, taken as controls. Significant differences with respect to controls were determined based on a non-parametric Kolmogorov–Smirnov test (** p* < 0.05; *** p* < 0.01) (Figure 5).

## 5. Conclusions

Composting is a natural process for recycling organic waste and involves the decomposition of organic matter by naturally occurring microorganisms. Starting materials commonly used in composting include municipal green waste (MGW) or agro-based waste. The effects of MGW composts mixed with biological control agents (BCAs) on plant parasitic nematodes have recently been reviewed. We tried to control RKNs using a food waste compost obtained by two electromechanical composters. The analysis of its organic matter components and, in particular, its high humification degree encouraged us to start a series of experiments to detect its performance both as a PG-promoter organic amendment and a tool of microbiome-assisted tntegrated pest management. We had already studied the suppressive effect of commercial AMF-containing products both on nematode and insect infestations. Most of the tested products were able to consistently control RKN infections, although they did not have a PGP effect on nematode-infected and uninfected plants. The relief of symptoms caused by these formulations balanced plant growth with the fitness costs caused by the diversion of energy metabolism from development and growth to defense. Therefore, all of the tested commercial microbiome-generating formulations could be indicated as bio-stimulants but not as bio-fertilizers. On the contrary, the compost tested in this study has shown both properties and can be used as a product that can join the markets of organic composted bio-fertilizers and that of BCAs. Future studies will focus on the possibilities of enhancing the performance of composts like FWC1 by mixing them with suitable AMF/microbe strains either in the production process or in soil treatments.

## Figures and Tables

**Figure 1 ijms-26-10606-f001:**
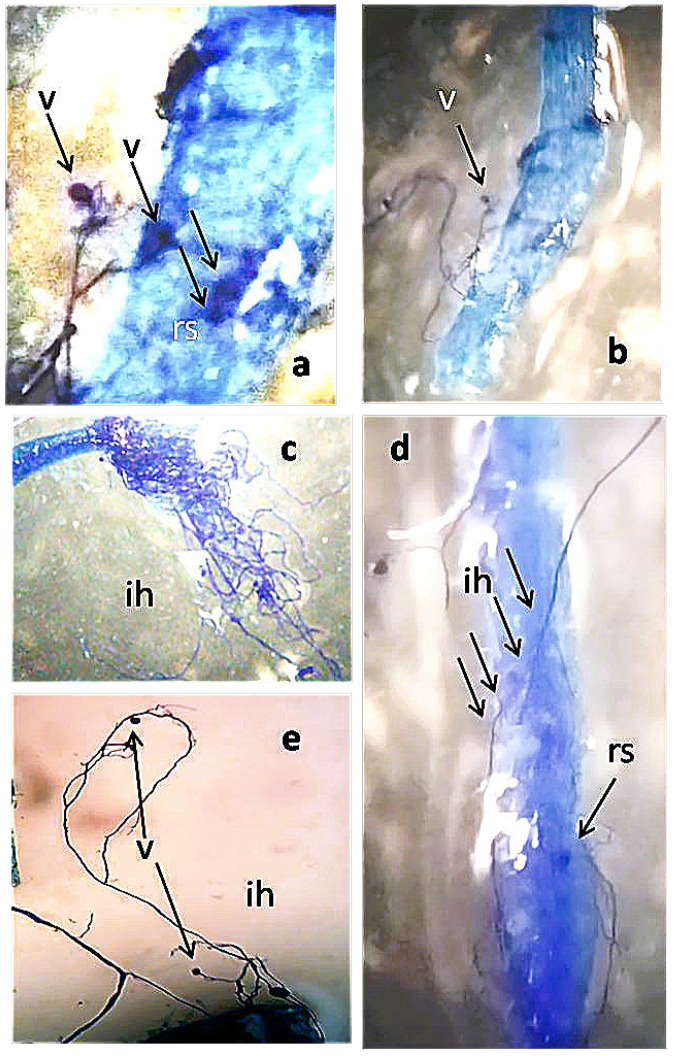
Tomato roots infected with arbuscular mycorrhizal fungi 1 month after FWC1 treatment, cleared and stained with KOH-lactophenol blue: (**a**) roots colonized by resting spores (rs) and vesicles (v) 100×, (**b**) 50×; (**c**,**d**) AMF-infected area and intra- and extra-radical mycelium (ih) with 1 resting spore × 50; (**e**) hyphae (ih) and vesicles emerging from roots 50×.

**Figure 2 ijms-26-10606-f002:**
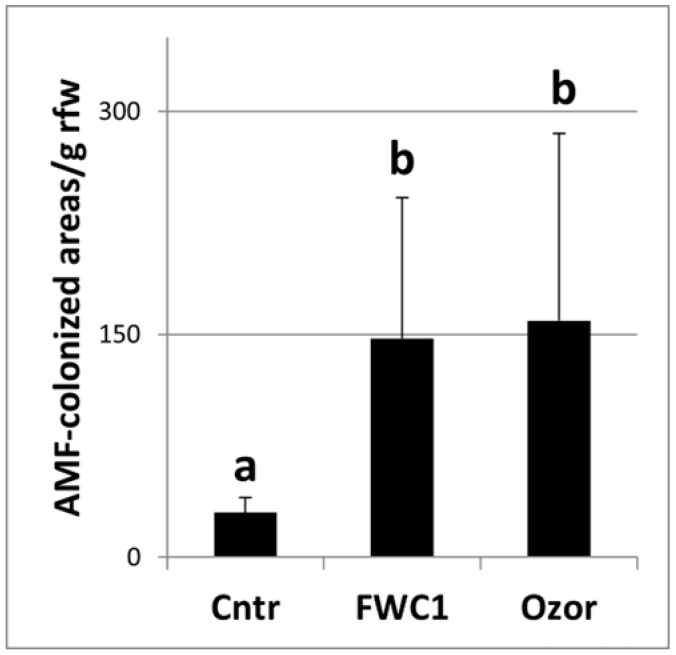
AMF-infected areas per g of root fresh weight (rfw) observed on KOH-lactophenol blue-treated roots under a stereoscope. Roots were collected 1 month after inoculation of plants with FWC1 and Ozor; another group of plants was left uninoculated (Cntr). Six different root samples coming from different plants were analyzed per treatment. Values are expressed as means (*n* = 6) ± standard deviations. Means were separated based on a Duncan’s Test, where different letters indicate significantly different means (significance level: 0.01).

**Figure 3 ijms-26-10606-f003:**
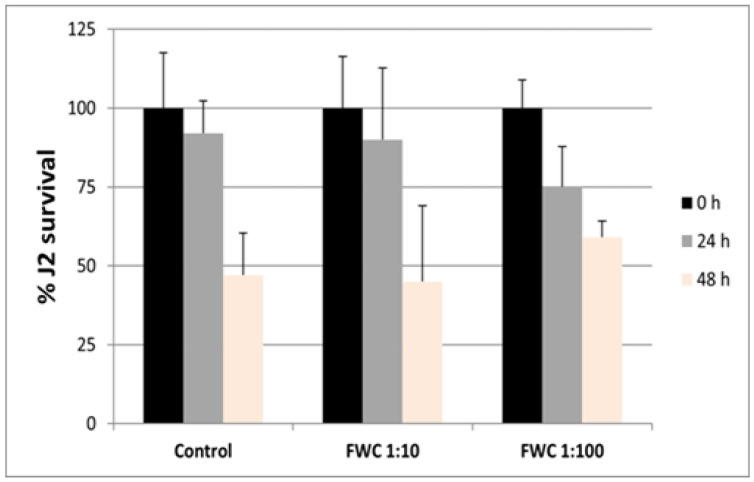
Percentages of J2 survival after 24–48 h incubation in diluted (1:10, 1:100) aqueous extracts of FWC1. Means ± standard deviations were obtained from 3 values, and there was an absence of significance differences between the means of survived control J2s and those of FWC1-incubated J2s at different incubation times, characterized by the same color. This was verified by a paired *t*-test.

**Figure 4 ijms-26-10606-f004:**
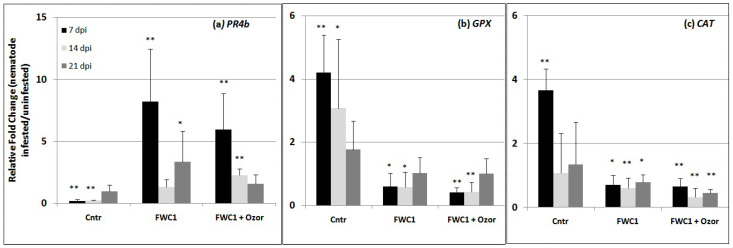
Expression of the genes encoding for pathogenesis-related protein 4b (PR4b) and for the enzymes glutathione peroxidase GPX and catalase CAT in roots of tomato plants untreated (Cntr) and treated with FWC1 and FWC1 mixed with minimal amounts of Ozor, detected using q-RT-PCR. Relative fold changes (the value 1 indicates no change) of nematode inoculated with respect to uninoculated plants were determined at 7, 14, 21 dpi. Data are the mean fold changes (*n* = 6) ± SD in gene transcript levels. Asterisks indicate that the mean fold change is significantly different from 1, as determined by the non-parametric Kolmogorov–Smirnov test (** p* < 0.05; *** p* < 0.01).

**Figure 5 ijms-26-10606-f005:**
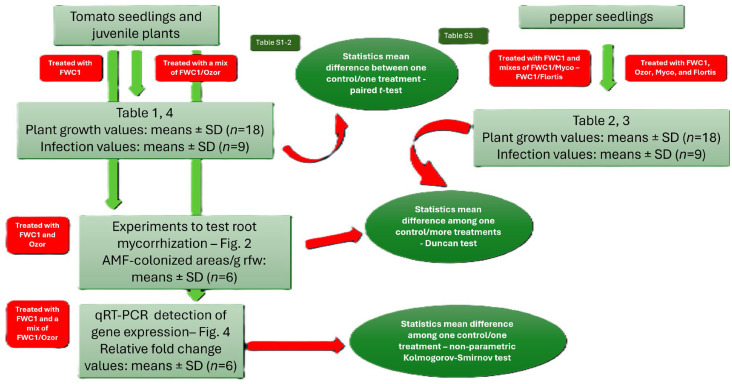
Flow chart of the experiments: plant species involved, treatments, mean values, replications, and statistics. Table 1, Table 2, Table 3 and Table 4, and Figure 2 and Figure 4 are shown above in the Section 2.

**Table 1 ijms-26-10606-t001:** Effect of FWC1 and FWC1 mixed with minimal amounts of Ozor on RKN-inoculated and uninoculated tomato plants at different ages. Plant growth was measured 50 days after treatment, including shoot height (SH in cm), shoot weight (SW in g), and root weight (RW, in g). Infection factors were detected 50 days after inoculation as egg masses per g root fresh weight (EMs g^−1^ rfw), sedentary forms per g root fresh weight (SFs g^−1^), female fecundity (FF), and reproduction potential (RP). Values are expressed as the means (*n* = 18, 9) ± standard deviations (SD). All the parameters of the treated plants were compared to those of untreated control plants (Cntr). Significant changes, according to a *t*-test (*p* < 0.05), are indicated by an asterisk. Significant differences between the treated and control plants (Cntr) are indicated in %.

	1Cntr ^a^	1FWC1 ^b^	2Cntr ^c^	2FWC1 ^d^	3Cntr ^c^	3FWC1 + Ozor ^e^
uninoculated						
SH	36 ± 12	45 ± 13 * (58)	59 ± 6	68 ± 12	51 ± 13	66 ± 11 * (29)
SW	10.9 ± 3.6	17.2 ± 3.0 * (57)	18 ± 6.7	21.9 ± 12.8 * (20)	18.7 ± 8.9	25.8 ± 14.3 * (38)
RW	1.0 ± 0.3	1.4 ± 0.4 * (40)	1.3 ± 0.7	1.8 ± 1.2 * (38)	1.3 ± 0.7	1.5 ± 0.8
inoculated						
SH	29 ± 7	33 ± 10	40 ± 9	39 ± 9	42 ± 10	48 ± 5
SW	7.5 ± 5.8	10.0 ± 6.9 * (33)	12.7 ± 4.7	12.7 ± 6.3	12.7 ± 4.7	17.2 ± 4.3 * (35)
RW	1.5 ± 1.2	1.7 ± 1.1	1.1 ± 0.4	1.2 ± 0.9	1.1 ± 0.4	1.4 ± 0.5 * (35)
EMs g^−1^	70 ± 33	35 ± 26 * (−51)	75 ± 38	56 ± 39 * (−26)	64 ± 41	52 ± 40 * (−18)
SFs g^−1^	104 ± 62	57 ± 50 * (−45)	108 ± 39	70 ± 37 (−35)	107 ± 38	84 ± 19 * (−22)
FF	214 ± 78	293 ± 122 * (37)	287 ± 107	457 ± 140 * (59)	--	--
RP	71 ± 24	51 ± 19 * (−28)	55 ± 14	53 ± 23	--	--

^a^ seedling (1.5–3.0 g); ^b^ FWC1 10 g kg^−1^ soil; ^c^ juvenile plants (5.0–7.0 g). ^d^ FWC1 30 g kg^−1^ soil; ^e^ FWC1 30 g kg^−1^ soil; Ozor 0.6 g kg^−1^ soil.

**Table 2 ijms-26-10606-t002:** Effects of FGW1 and FGW1 mixed with minimal amounts of 2 microbiome-generating formulations (Myco, Flortis) on pepper seedlings. Plant growth was detected 50 days after treatment as shoot height (SH in cm), shoot weight (SW in g), and root weight (RW, in g); infection factors were detected 50 days after inoculation as egg mass per g root fresh weight (EMs g^−1^ rfw), sedentary forms per g root fresh weight (SFs g^−1^), female fecundity (FF), and reproduction potential (RP). Values are expressed as the means (*n* = 18, 9) ± standard deviations (SD). Significant changes according to a Duncan test (*p* < 0.05) are indicated by different letters in each row.

	Cntr	FWC1	FWC1 + Myco	FWC1 + Flortis
SH	29 ± 6 ^a^	27 ± 5 ^a^	30 ± 2 ^a^	29 ± 3 ^a^
SW	8.5 ± 3.9 ^c^	12.0 ± 5.0 ^a^	10.5 ± 2.3 ^b^	9.0 ± 1.8 ^bc^
RW	1.7 ± 0.9 ^b^	2.5 ± 1.1 ^a^	2.7 ± 0.4 ^a^	1.9 ± 0.7 ^b^
EMs g^−1^	145 ± 61 ^a^	84 ± 21 ^b^	59 ± 30 ^c^	105 ± 15 ^b^
SFs g^−1^	253 ± 107 ^a^	140 ± 52 ^b^	141 ± 31 ^b^	166 ± 46 ^b^
FF	492 ± 91 ^b^	586 ± 209 ^a^	588 ± 159 ^a^	341 ± 46 ^c^
RP	171 ± 31 ^a^	109 ± 33 ^b^	97 ± 43 ^b^	98 ± 46 ^b^

**Table 3 ijms-26-10606-t003:** Comparison between FWC1 treatment and treatments with different AMF formulations (Ozor, Myco, Flortis) at the most effective doses on RKN-inoculated pepper seedlings. Plant growth was detected 50 days after treatment as shoot height (SH in cm), shoot weight (SW in g), and root weight (RW, in g); infection factors were detected 50 days after inoculation as egg mass per g root fresh weight (EMs g^−1^ rfw), sedentary forms per g root fresh weight (SFs g^−1^), female fecundity (FF), and reproduction potential (RP). All the parameters of the treated plants were compared to those of plants left untreated, as controls (Cntr). Significant changes, according to a Duncan test (*p* < 0.05), are indicated by different letters in each row.

	Cntr	+FWC1	+Ozor	+Myco	+Flortis
SH	29 ± 5 ^a^	30 ± 10 ^a^	27 ± 4 ^a^	30 ± 15 ^a^	29 ± 4 ^a^
SW	7.6 ± 2.4 ^b^	10.6 ± 4.3 ^a^	8.3 ± 5.3 ^b^	8.5 ± 2.6 ^b^	7.6 ± 2.8 ^b^
RW	1.5 ± 0.9 ^b^	2.2 ± 1.0 ^a^	1.5 ± 0.8 ^b^	1.6 ± 1.3 ^b^	1.4 ± 0.8 ^b^
EMs g^−1^	145 ± 50 ^a^	100 ± 27 ^b^	106 ± 32 ^b^	96 ± 53 ^b^	93 ± 53 ^b^
SFs g^−1^	215 ± 85 ^a^	162 ± 61 ^b^	118 ± 43 ^c^	155 ± 62 ^b^	182 ± 65 ^ab^
FF	502 ± 138 ^a^	490 ± 177 ^a^	547 ± 70 ^a^	522 ± 112 ^a^	396 ± 90 ^b^
RP	147 ± 45 ^a^	117 ± 28 ^b^	97 ± 23 ^b^	113 ± 65 ^b^	104 ± 46 ^b^

**Table 4 ijms-26-10606-t004:** Duration of the effect of FCW1 on subsequent crop seasons. Tomato was planted in the soil of eggplants treated 2 months earlier. Plant growth was detected as shoot height (SH in cm), shoot weight (SW in g), and root weight (RW, in g); infection factors were detected 50 days after inoculation as egg mass per g root fresh weight (EMs g^−1^ rfw), sedentary forms per g root fresh weight (SFs g^−1^), female fecundity (FF), and reproduction potential (RP). All the parameters of the treated plants were compared to those of plants left untreated, as controls (Cntr). Significant changes, according to a *t*-test (*p* < 0.05), are indicated by an asterisk. Significant differences in the treated plants compared to the control plants (Cntr) are indicated in %.

	1st Crop: Eggplant		2nd Crop: Tomato	
	Cntr	FWC1	Cntr	FWC1
SH	40 ± 7	41 ± 3	59 ± 7	63 ± 8
SW	11.3 ± 6.5	13.5 ± 7.9 * (19)	29.1 ± 10.3	28.8 ± 6.9
RW	1.9 ± 1.6	1.9 ± 1.4	2.5 ± 0.8	1.5 ± 0.6 * (−40)
EMs/g	45 ± 23	15 ± 6 * (−67)	31 ± 13	21 ± 15 * (−32)
SFs/g	120 ± 48	52 ± 20 * (−57)	59 ± 35	59 ± 12
FF	234 ± 63	483 ± 116 *(106)	160 ± 72	122 ± 57
RP	39 ± 28	25 ± 5 * (−36)	21 ± 2	15 ± 5 * (−29)

**Table 5 ijms-26-10606-t005:** Composition in micro-elements, chemo-physical parameters, organic matter content, and elemental analysis of FWC1.

Micro-Elements				
Name	Unit	Legal Limits	Value	±Ue
Cd	mg kg^−1^	1.5	0.382	0.074
Cr (VI)	mg kg^−1^	0.5	0.592	0.019
Mn	mg kg^−1^		478	13
Hg	mg kg^−1^	1.5	0.154	0.008
Ni	mg kg^−1^	100	64.5	14.2
Pb	mg kg^−1^	140	31.4	0.7
Cu	mg kg^−1^	230	143	14
Zn	mg kg^−1^	500	247	23
Ca	%		6.39	0.14
Fe	%		1.50	0.15
P	%		0.373	0.014
Mg	%		0.547	0.023
K	%		1.97	0.03
Organic Matter				
TEC	%		16.9	0.5
HA + FA	%		13.8	0.3
DH	%		82	
Chemo-Physical Parameters				
Name	Unit	Value Min	Value Max	
pH		7.5	7.7	
RH	%	15	18	
Elemental Analysis				
Name	Unit	Average Value	C:N	
Carbon	%	27.23	13	
Nitrogen	%	2.07		

## Data Availability

The original contributions presented in this study are included in the article. Further inquiries can be directed to the corresponding author.

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
