# Peer review of "Food Waste Compost as a Tool of Microbiome-Assisted Agri-Culture for Sustainable Protection of Vegetable Crops Against Soil-Borne Parasites"

_ijms, 2025, doi:10.3390/ijms262110606_

Round 1
Reviewer 1 Report (New Reviewer)
Comments and Suggestions for Authors
The manuscript investigates the use of a specific food waste compost (FWC1) as a dual-purpose bio-fertilizer and bio-stimulant against root-knot nematodes (RKNs). The results suggest that the sustainably produced food waste compost can effectively promote plant growth and induce resistance. The topic is of interest, the experiments are generally comprehensive, and the findings are significant for the fields of sustainable agriculture and plant-microbe interactions. Some points should be improved as following.
L97, "It has long known" should be "It has long been known".
Table 1, The column headings in Table 1 are confusing. It appears Cntra, FWC1b, Cntrc, FWC1d, Cntrc, FWC1+Ozore represent three distinct experiments (seedlings, juvenile, juvenile+mix). This should be made visually clearer, perhaps with spanning headers. Also, the parameter SFs in juvenile plants 70±37e, it sounds an error.
Figure 1, Please ensure they are of high resolution (a, b, d), and explain the ih.
Figures 2-4, The quality of the figures should be improved. Please provide the statistical annotations in Figure 2 and 3.
Figure 2 shows that FWC1 application leads to root colonization, but it does not prove that the colonizing AMF originated from FWC1 and not from the native soil or other sources.
Figure 4, The y-axis label "Relative Fold Change (nematode infected/uninfected)" is misinterpreted. It is also unclear what the "value 1" baseline represents in the context of the different treatment groups.
L355-360, This sentence is grammatically flawed.
Table 6 and Figure 5 can be used as the supplementary materials. And Figure 5 should be cited in the text.
Author Response
The manuscript investigates the use of a specific food waste compost (FWC1) as a dual-purpose bio-fertilizer and bio-stimulant against root-knot nematodes (RKNs). The results suggest that the sustainably produced food waste compost can effectively promote plant growth and induce resistance. The topic is of interest, the experiments are generally comprehensive, and the findings are significant for the fields of sustainable agriculture and plant-microbe interactions. Some points should be improved as following.
L97, "It has long known" should be "It has long been known".
ok
Table 1, The column headings in Table 1 are confusing. It appears Cntra, FWC1b, Cntrc, FWC1d, Cntrc, FWC1+Ozore represent three distinct experiments (seedlings, juvenile, juvenile+mix). This should be made visually clearer, perhaps with spanning headers.
We gave a number to each experiment to make visible that Table 1 deals with 3 different experiments
Also, the parameter SFs in juvenile plants 70±37e, it sounds an error.
It was just a typo, we deleted the e
Figure 1, Please ensure they are of high resolution (a, b, d), and explain the ih.
The figure has been improved and ih explained in the revised version
Figures 2-4, The quality of the figures should be improved. Please provide the statistical annotations in Figure 2 and 3.
In Fig. 2 statistical annotations are already described in the legend. We added statistical annotations in the legend of Fig. 3
Figure 2 shows that FWC1 application leads to root colonization, but it does not prove that the colonizing AMF originated from FWC1 and not from the native soil or other sources.
Plants grown in native soil are described as CNTR in the figure: their roots show significantly lower numbers of AMF-colonized areas per g of root fresh weight with respect to roots of plants treated with FWC1 and Ozor
Figure 4, The y-axis label "Relative Fold Change (nematode infected/uninfected)" is misinterpreted. It is also unclear what the "value 1" baseline represents in the context of the different treatment groups.
In the legend of Fig. 4 is written: “Relative fold changes (the value 1 indicates no change) of nematode inoculated with respect to not inoculated plants were determined at 7, 14, 21 dpi. Data are the mean fold changes (n=6) ± SD in gene transcript levels. Asterisks indicate that the mean fold change is significantly different from 1 as determined by the non-parametric Kolmogorov-Smirnov test (*p<0.05; **p<0.01)”
When the ratio between gene transcript levels, determined as 2-∆∆Ct by q-RT-PCR, of the inoculated sample and the respective not inoculated sample is 1, it means that such levels are not significantly different. Values <1 mean that such levels in inoculated samples are lower than those in not inoculated samples; values >1 mean that such levels in inoculated samples are higher than those in not inoculated samples. See for instance:
Molinari S, Fanelli E., Leonetti P. 2014. Expression of tomato salicylic acid (SA)-responsive pathogenesis-related genes in Mi-1-mediated and SA-induced resistance to root-knot nematodes. Molecular Plant Pathology, 15, 255–264 DOI: 10.1111/mpp.12085
Molinari S., Leonetti P. (2019). Bio-control agents activate plant immune response and prime susceptible tomato against root-knot nematodes. PloS One, 14(12): e0213230.
https://doi.org/10.1371/journal.pone.0213230
Molinari S., Akbarimotlagh M. and Leonetti P. (2022). Tomato root colonization by exogenously inoculated arbuscular mycorrhizal fungi induces resistance against root-knot nematodes in a dose-dependent manner. Int. J. Mol. Sci. 23, 8920 doi: 10.3390/ijms23168920
L355-360, This sentence is grammatically flawed.
It has been changed
Table 6 and Figure 5 can be used as the supplementary materials. And Figure 5 should be cited in the text.
Ok, Table 6 has been removed from the text and added as a supplementary Table (Table S4). Figure 5 has been requested in the main text by another reviewer. It has been cited in the text of the revised version

Reviewer 2 Report (New Reviewer)
Comments and Suggestions for Authors
This study evaluates the performance of a compost derived from canteen food waste (Food Waste Compost 1, FWC1) as a growth-promoting organic amendment and a resistance inducer against root-knot nematodes.
Overall, the manuscript is informative. The methods are sound, and the results appear reliable. Below are some suggestions for improvement:
It is recommended that the authors consider conducting an experiment to measure the germination index (GI) of FWC1, which would enhance the toxicity analysis. It was surprising that such an experiment was not included. If there is a valid reason for its omission, it should be discussed in detail.
In the Introduction, a more comprehensive literature review is needed. The significance of FWC1 as a novel contribution to the field should be clearly highlighted. Existing knowledge gaps should be addressed, and the objectives of the study should be clearly stated.
Kindly add a measurement scale to the microscope images in Figure 1 to facilitate precise evaluation of object size. The zoom value in the figure caption alone may not adequately convey the scale.
In conclusion, the manuscript requires further revision before it can be considered for publication.
Author Response
This study evaluates the performance of a compost derived from canteen food waste (Food Waste Compost 1, FWC1) as a growth-promoting organic amendment and a resistance inducer against root-knot nematodes.
Overall, the manuscript is informative. The methods are sound, and the results appear reliable. Below are some suggestions for improvement:
It is recommended that the authors consider conducting an experiment to measure the germination index (GI) of FWC1, which would enhance the toxicity analysis. It was surprising that such an experiment was not included. If there is a valid reason for its omission, it should be discussed in detail.
Germination index is a test made on seeds put in the compost to determine the germination % and the length of roots with respect of seeds put in a control substrate. First, every biological treatment, if the added compound is provided in excess, may be toxic to plants. Therefore, compost under our experimental conditions is added at a rate of 6-30 g/kg soil, depending on the age of plants. Higher rates may be toxic to plants. Therefore, toxicity depends on the amount of compost we provide to plants, and it is not a characteristic of a specific compost or any other microorganism formulation. At the amounts of compost we provide to soil, treated plants grow much better than not treated plants, as attested by Table 1. Of course, compost, at the used rates, IS NOT toxic to plants but is a good bio-fertilizer, although also good bio-fertilizers should not be used in excess. What was important to test was the toxicity of the compost, used at plant-friendly rates, on living nematode juveniles. Many products may be toxic to nematode juveniles and NOT to plants (consider the use of any nematicide, for instance), at specific concentrations. Data shown on Figure 3 ruled out this occurrence. Of course, in vitro tests can give an idea on the rates of J2s mortality caused by a given compound, but they cannot predict a reduction of the parasitism potential, i.e. movement impairing, that can occur in vivo; this is clearly stated in the text. Therefore, toxicity to plants and to nematode worms may have different factors.
In the Introduction, a more comprehensive literature review is needed. The significance of FWC1 as a novel contribution to the field should be clearly highlighted. Existing knowledge gaps should be addressed, and the objectives of the study should be clearly stated.
That FWC1 may be a novel contribution to the field can be attested after that we have shown it by our data in the Result Section. IN fact, we have stressed the novelty of its potential use for a possible joint market of biofertilizers and biostimulants in the Conclusion Section, where the objectives of the study are clearly addressed. State-of-the-art and knowledge gaps are, in our opinion, already indicated in the extensive list of references dedicated to the Introduction Section [1-15 out of 31 total]. If the referee may kindly suggest additional key reports, we would be happy to add them to our Introduction.
Kindly add a measurement scale to the microscope images in Figure 1 to facilitate precise evaluation of object size. The zoom value in the figure caption alone may not adequately convey the scale.
Usually, we use the zoom values for stereoscope measurements because we set it on the stereoscope. This has long been common in such types of measurements, see Phillips and Hayman, Transactions British Mycological Society 1970, 55, 158-161. Although we can understand your opinion, we do not have the means to add a measurement scale and do not think this may be essential for the understanding of the figure.
In conclusion, the manuscript requires further revision before it can be considered for publication.

Round 2
Reviewer 2 Report (New Reviewer)
Comments and Suggestions for Authors
This is the second review round for manuscript ijms-3930404. The reviewer notes that the authors’ responses were surprisingly dismissive, suggesting they took the comments lightly and did not engage with sufficient seriousness.
The response clearly indicates that the authors are unfamiliar with the concept of the Germination Index (GI). The phrase “seeds put in the compost” is inaccurate. GI is measured using a compost-amended substrate or compost extract, not by directly burying seeds in raw compost. The authors are advised to first understand what GI is and how it is measured. It is recommended that they conduct an additional experiment and familiarize themselves with relevant factors such as seed species, extract preparation or substrate composition, incubation period, and the formula used to calculate GI.
The statement that toxicity “is not a characteristic of a specific compost” is incorrect. Phytotoxicity varies depending on the dose, compost composition, and maturity. Therefore, some composts may be inherently more phytotoxic than others.
The range “6–30 g/kg soil” is presented without any source or justification. The authors should clarify whether this range is based on literature, preliminary trials, or specific crop or plant age requirements. Please explain the rationale behind selecting these application rates.
Author Response
please, see the attachment

This manuscript is a resubmission of an earlier submission. The following is a list of the peer review reports and author responses from that submission.
Round 1
Reviewer 1 Report
Comments and Suggestions for Authors
In this manuscript, Paola Leonetti and colleagues analyzed the effect of plant defense against root-knot nematodes. Although the work appears to have been carried out with care, in my judgment, your manuscript seems not to be associated with "molecules" and unsuitable for IJMS.
I am sorry that I cannot be more positive on this occasion. However, I think your paper may be suitable for the MDPI journals, Agronomy or Agriculture.
Author Response
In this manuscript, Paola Leonetti and colleagues analyzed the effect of plant defense against root-knot nematodes. Although the work appears to have been carried out with care, in my judgment, your manuscript seems not to be associated with "molecules" and unsuitable for IJMS.
I am sorry that I cannot be more positive on this occasion. However, I think your paper may be suitable for the MDPI journals, Agronomy or Agriculture.
Transcripts of mRNA detected by qRT-PCR are MOLECULES, if we are not wrong – we have published several papers with these type of data on IJMS
Reviewer 2 Report
Comments and Suggestions for Authors
Authors provided a strong evidence for the bio-control of RKNs by food waste compost in simulation experiments. Specific issues should be addressed before being accepted.
Authors should check the manuscript carefully to avoid the mistakes in format.
1) Title
-change “Low Scale Composting” to “Food Waste Compost”
2) Abstract
-line 17: change to “root-knot nematodes (RKNs)”
-line 23: change to “arbuscular mycorrhizal fungi (AMF)”
-line 27: change to “FWC1 primes”
3) Introduction
-line 69-85: to condense the description of the mechanism of RKNs on plant growth.
4) Materials and Methods
-to give the schematic diagram of the composter to reduce the description of this device.
-table 5: the decimal point was written as a comma, such as 1,97 in K.
-line 544: the extract of FWC1 at 0.3 g/mL was too low.
5) Results and Discussion
-Table 2: to label the maximum value with the letter a.
-table 1 and figure 2: Ozor can be replaced with AMF.
6) Conclusions
-there is no need to cite the references in the conclusion section.
7) References
-to increase the citations of relevant literature on this topic for the past three years.
Author Response
Rev. 2
Authors provided a strong evidence for the bio-control of RKNs by food waste compost in simulation experiments. Specific issues should be addressed before being accepted.
Authors should check the manuscript carefully to avoid the mistakes in format.
1) Title
-change “Low Scale Composting” to “Food Waste Compost”
ok
2) Abstract
-line 17: change to “root-knot nematodes (RKNs)”
-line 23: change to “arbuscular mycorrhizal fungi (AMF)”
-line 27: change to “FWC1 primes”
done
3) Introduction
-line 69-85: to condense the description of the mechanism of RKNs on plant growth.
done
4) Materials and Methods
-to give the schematic diagram of the composter to reduce the description of this device.
The schematic diagram is shown on the indicated web site https://www.compostkmzero.it/index.php/modelli - We highlighted it in the revised version
-table 5: the decimal point was written as a comma, such as 1,97 in K.
done
-line 544 (554): the extract of FWC1 at 0.3 g/mL was too low
We are at the limits of FWC1 solubility – in experiments with plants 0.07 g/ml FWC1 is added at the highest, that’s why we diluted the extract
5) Results and Discussion
-Table 2: to label the maximum value with the letter a.
done
-table 1 and figure 2: Ozor can be replaced with AMF
Ozor indicates the amount of AMF added
6) Conclusions
-there is no need to cite the references in the conclusion section.
Ok, we deleted the citations
7) References
-to increase the citations of relevant literature on this topic for the past three years.
We added recent literature in References
Reviewer 3 Report
Comments and Suggestions for Authors
What was the significance of choosing the food waste from that particular source/ kitchen?
What was the justification for using kitchen waste? And specifically that particular kitchen?
How constant, stable in components would that waste be along days, weeks, months, seasons, and at other locations?
Note the comments raised in the attached file.

Author Response
Rev. 3
The Title
• “low scale”. How exact is this description? Scientifically, it is not. Where does it stand
against high and medium-scale composting? What was the scale of measuring,
determining the levels of composting?
The level of “scale” production is industrial production in terms of tons and correlative market proposal (lots of web sites sell composts in tons) considered as “high-scale production” – our production is NOT industrial and in terms of kgs., therefore, the terms low scale production (decentralized composting) sounds rationale. Anyway, we changed the Title and removed the "low scale".
“a Tool of Microbiome-Assisted Agriculture”!!??
Yes, Microbiome-Assisted Agriculture implies the use of composts as a tool, very simple
The materials used in this work were obtained from a kitchen at a specific time and
composted. Then it was applied to plants treated with commercial Microbiomes
No, compost was applied AS SUCH first, THEN it was tested mixed with commercial microbe formulations, therefore the suggested title below would not be correct – all data mostly focus on treatments with compost, molecular ones included.
• So, the current title does not describe, nor does it introduce this work. A closer and more suitable title may be phrased as follows; Composted kitchen Food Waste from the Casaccia Research Centre, Department for Sustainability of ENEA, Rome, Italy enhance the effects of microbiome treatments and the tolerance to root knot nematode on tomato and pepper plants
all food comes from a “kitchen”, hopefully, except fresh food that contributed to the raw material for compost production; our compost comes from not a simple "kitchen" but from a canteen with 250/300 meals a day – Materials and Methods are normally not included in the Title – we accepted the corrections of a different reviewer and titled as: Food Waste Compost as a Tool of Microbiome-Assisted Agri-culture for Sustainable Protection of Vegetable Crops Against Soil-Borne Parasites
Lines # 421-42/5 “For this work, the EC was loaded daily (from Monday to Friday) with 15–20 kg of biowaste and 2.2–5.5 kg of bulking agent (pruning of Arundo donax canes, shredded in pieces of 2- 5 cm) which represented 15–20% wt of the food waste) for 60 days and completely emp?ed on the 91st day to ensure an average residence ?me for all the organic ma?er. At the beginning of the experiment, 25 kg of mature compost was uniformly distributed inside the EC chamber as an inoculum of microorganisms”
Lines # 246-249 (428-430) “The biowaste used consisted of leftovers and kitchen scraps from the research center canteen. The source-separated origin of biowaste and the method of collection ensured a very low amount of foreign materials”
• The only ingredient mentioned specifically is the Arundo donax, but no justification is given for why it is used.
Line 424: Arundo donax Bulking agent: In a composting context, a bulking agent is a material added to the compost pile to improve its structure and aeration, thus promoting the decomposition process. This material, usually rich in carbon, helps balance the compost's composition, preventing compaction and improving drainage.
• The whole section needs to be turned around
Why?
• First, what were the ingredients that went into the composter?
• “consisted of leftovers and kitchen scraps from the research center canteen.” This makes all of this work and results are limited to the materials collected from that specific kitchen at that specific ?me. Hence, present findings cannot be universally expected from using “le?overs and kitchen scraps” from other kitchens at other locations in the country, and more
is the referee saying that composts should be “universally the same”? we named THIS COMPOST FWC1 and wrote a paper on it because it comes from a specific canteen, specific bulking agents, a specific processing, etc., and FOR THESE REASONS FWC1 is so specific – we would be a little disappointed if the quality of our compost were “universal” – we really don’t get what the referee wants to say
“The source-separated origin of biowaste and the method of collection ensured a very low amount of foreign materials.” This is a very ambiguous, non-specific, and nonscientific statement.
Ok, we deleted the non-scientific statement from the text -however, foreign materials include all that is not compostable like plastic, metals, glass, etc.; in our case, separation of food waste from foreign materials is manually done at the trays level, therefore, the definition of "source-separated biowaste" indicates a high quality Food Waste
• What is meant by “The source-separated origin of biowaste” !!?? How constant can that be? How repeatable could that be?
We deleted the sentence
• The following is a non-scientific research experimental design, by saying “ensured a very low amount of foreign materials”!! What were those “foreign materials”? How were they avoided? So. That ingredient contained unknown amounts of “foreign materials” every time it was used
Lines 421-422, The input capacity of the EC is 25–30 kg day and the filling grade is 60–70%, which corresponds to a storage capacity of 350–450 kg
“Finally, the mature compost”. This is a non-scientific term. What is “mature compost” in this specific case? On what basis, criteria, and other measurable parameters, was the maturity determined?
It is absolutely scientific term, as follows: If composting is done correctly, the resulting compost can be used after 4–6 months, but it is typically mature after 8–10 months. The compost's fertilizing capacity increases significantly as it matures - mature compost can also be said "cured compost"
• So, the above comments are not for some semantic and language criticism. • Not clearly defined, described, and executed experimental materials and their variables cast doubt on the whole work and its results. It will also be irreparable.
we showed that the adopted terms are commonly used in the scientific literature of the topic and that the ways of processing are clearly described. We deleted the sentence that Rev. 3 considered to be “very ambiguous, non-specific, and nonscientific statement”
Figure 1. Tomato roots infected with arbuscular mycorrhizal fungi 1 month after FWC1 treatment, 218 cleared and stained with KOH-lactophenol blue: (a) roots invaded by resting spores (rs) and vesicles (v) x 100, (b) x 50; (c-d) AMF-infected area and intra- and extra-radical mycelium (I-ERM) with 1 res?ng spore x 50; (e) hyphae and vesicles emerging from roots x 50.
• The whole figures are of a low quality, especially figures a, b, c and d. Those show no rot tissues and out of contrast.
Why should figures show rot tissues? AMF are symbionts not pathogenic. Figs 1b and 1d had their contrast lowered to highlight a vescicle and hyphae inside the root, respectively
“invaded by resting spores (rs) and vesicles.” This is not mycologically sound descrip?on. Resting spores do not invade plant tissues. Vesicles are fungal structures developed inside the plant tissues.
Ok, invaded has been substituted by “colonized” in the revised version
Table 3. Comparison between FWC1 treatment and treatments with different AMF formulations 184 (Ozor, Myco, Flortis) at the most effective doses on RKN-inoculated pepper plantlets. There are three microbiome treatments and a different crop.
Tab. 3 shows the effects of FWC1, Ozor, Myco, and Flortis treatments on pepper seedlings, we don’t get the comment
Table 4. Duration of the effect of FGW1 on subsequent crop seasons. Tomato was planted in the soil 203 of egg plants treated 2 months earlier. Plant growth was detected as shoot height (SH in cm), shoot weight (SW in g), and root weight (RW, in g); infection factors were detected 50 days after inocula?on as egg masses per g root fresh weight (EMs g-1 rfw), sedentary forms per g root fresh weight (SFs g-1), female fecundity (FF), and reproduc?on poten?al (RP). All theparameters of treated plants were referred to those of plants left untreated, as controls (Cntr). Significant changes, according to a t-test (P<0.05), are indicated by an asterisk. Significant difference in treated with respect to control 209 plants (Cntr) is indicated in %. This table contains readings, data obtained using plants of different ages, some were 50 days old, and others??? days old.
Of course not, egg plants and tomato were the same age – tomato was seeded some time after trial with egg plants had been started
There were two crops of this crop, which were not clearly described. “female fecundity (FF), and reproduction on potential (RP).” How were the numerical values of these parameters produced?
It is written in the Materials and Methods Section lines 531-536
Reviewer 4 Report
Comments and Suggestions for Authors
Review on “Low scale composting as a tool of Microbiome-Assisted Agriculture for sustainable protection of vegetable crops against soil-borne parasites” for manuscript ID ijms-3803212
In this manuscript the authors show the importance of sustainable agriculture practices, emphasizing the need for eco-friendly alternatives to chemical fertilizers and pesticides. They discuss the role of organic amendments, such as composts, in improving soil health, promoting plant growth, and suppressing soilborne pests, including nematodes. Moreover, the Introduction section underscores the potential of plant growth-promoting microorganisms, particularly AMF, to enhance plant resistance and nutrient uptake in an environmentally sustainable manner.
Despite the importance of considered topic, the manuscript has several flaws. The introduction section lacks the current state of knowledge about practices using compost against pests, particularly PPNs. The papers like the following could help to improve the intro:
Karanastasi E, Kotsantonis V, Pantelides IS. Compost-Derived Bacterial Communities Offer Promise as Biocontrol Agents against Meloidogyne javanica and Promote Plant Growth in Tomato. Agriculture. 2024; 14(6):891. https://doi.org/10.3390/agriculture14060891
Major points:
- Taxonomic Nomenclature
The manuscript mentions several organisms (e.g., pepper, eggplant, nematodes) without providing their scientific names. According to scientific conventions, the Latin binomial should be included at the first mention of each species. For example:
- Pepper — Capsicum annuum(or other species, if applicable)
- Eggplant — Solanum melongena
- Nematodes — The species/strain should be specified if possible (e.g., Meloidogyne incognitafor root-knot nematodes). If unknown, state "soil-dwelling nematodes (unspecified species).".
Without this information, the reproducibility of the study is compromised.
- Inconsistent Abbreviations
- The abbreviation "FWC1" is sometimes written as "FCW1" or "FWG1" (Table 1, line 166). This inconsistency must be corrected. Multiple abbreviations (e.g., HR, I-ERM, ROS, HA, FA, DH) are introduced but used only once, reducing readability. I would recommend to remove unnecessary abbreviations and use full terms, or keep only those repeated frequently (≥3–5 times).
- The abbreviation "PGP" (PGP traits, PGP effect) is not explained.
- Experimental Design and Presentation
The experiment is extremely difficult to interpret due to the inconsistency of its implementation and presentation of the results. Thus, the authors use different doses of compost for eggplant (possibly 6 g kg-1), pepper (6 g kg-1, 10 g kg-1) and tomato (10 g kg-1, 30 g kg-1) and explain this by the fact that they considered these doses to be effective. However, it is unclear how the effective dose was determined.
There is very little information about the experiment with eggplant plants. However, it is clear that preparations containing glomus fungi were not used in the experiments with eggplants.
The experiment with tomato plants is more complete. It contains information about plants of different ages and also about the effect of adding FCW1 together with Ozor on nematode infestation. However, there is no data on the effect of Ozor on the tomato plant and pathogenic nematodes by itself.
Then we read about the experiment with pepper plants and see that a special experiment was conducted for pepper plants with Myco and Flortis preparations containing glomus fungi. Surprisingly, the authors did not include Ozor in this experiment, despite the fact that in the next experiment with pepper plants, mixtures of Ozor, Myco and Flortis preparations with the studied FCW1 were used .
As a result, it is difficult for the reader to interpret the data, because the work is very hard to assemble into a single picture and looks more like a patchwork.
The problem is complicated by the fact that when compiling tables 1, 3, 4, some statistical methods were used (comparison only with the control), while in table 2, others were used (comparison of data from columns in pairs).
In order to make it easier for the reader to read the article, the structure of the work would be changed. Thus, it is necessary to divide the work into parts. The first part should contain experiments with eggplant, tomato and pepper plants, which describe the effect of using FCW1 compared to control plants. The second part should contain data on the effect of preparations with glomus fungi on their own and in combination with FCW1. Here, you can also provide illustrations with microphotographs.
The third part is needed to combine the data from parts one and two using statistical methods. It would also be appropriate to provide data on gene expression here (Figure 4), and in conclusion, add the information described in Figure 3:
- Effects of FCW1 alone(eggplant, tomato, pepper vs. controls).
- Effects of Glomus fungi preparations(Myco, Flortis, Ozor ± FCW1), with microphotographs.
- Integrated analysiscombining Parts 1–2, including gene expression (Fig. 4) and the information described in Figure 3.
Minor points:
Misleading Terminology
The term «adult plants» for tomato plants weighing 5–7 g is misleading, as this stage typically represents the juvenile vegetative phase, not reproductive maturity. I recommend using «juvenile plants» or «vegetative-stage plants» instead. Similarly, «plantlets» could be replaced with «seedlings» if referring to very young plants (1.5–2 g) with cotyledons or early true leaves.
Formatting Issues in Tables
- Table 2: The first column is too narrow, causing unit labels (e.g., "g⁻¹") to break across lines.
- Table 3: In the "+FCW1" column, some values repeat those in Table 2 (e.g. 84±21 (-42), 140±52 (-35), 586±209, 109±33 (-26)), while others do not (e.g. 30±10, 10.6±4.3 (40), 2.2±1.0 (51)). This inconsistency needs clarification.

Author Response
Rev. 4
Review on “Low scale composting as a tool of Microbiome-Assisted Agriculture for sustainable protection of vegetable crops against soil-borne parasites” for manuscript ID ijms-3803212
In this manuscript the authors show the importance of sustainable agriculture practices, emphasizing the need for eco-friendly alternatives to chemical fertilizers and pesticides. They discuss the role of organic amendments, such as composts, in improving soil health, promoting plant growth, and suppressing soilborne pests, including nematodes. Moreover, the Introduction section underscores the potential of plant growth-promoting microorganisms, particularly AMF, to enhance plant resistance and nutrient uptake in an environmentally sustainable manner.
Despite the importance of considered topic, the manuscript has several flaws. The introduction section lacks the current state of knowledge about practices using compost against pests, particularly PPNs.
The papers like the following could help to improve the intro:
Karanastasi E, Kotsantonis V, Pantelides IS. Compost-Derived Bacterial Communities Offer Promise as Biocontrol Agents against Meloidogyne javanica and Promote Plant Growth in Tomato. Agriculture. 2024; 14(6):891. https://doi.org/10.3390/agriculture14060891
Ok, we introduced this reference in the revised version because very recent, although we talked about “practices using compost against pests, particularly PPNs”, mentioning at least 3 important and extensive reports –
Reference n. 2 Bogdányi, F.T.; Pullai, K.B.; Doshi, P.; Erdős, E.; Gilián, L.D.; Lajos, K.; Leonetti, P.; Nagy, P.I.; Pantaleo, V.; Petrikovszki, R.; et al.. Composted Municipal Green Waste Infused with Biocontrol Agents to Control Plant Parasitic Nematodes—A Review. Microorganisms 2021, 13, 2130.
Reference n. 19 Daramola, F.Y.; Orisajo, S.B.; Osemwegie, O.O. Nematode Management Prospects in Composting. In: Sustainable Management of Nematodes in Agriculture Vol.1: Organic Management; Chaudhary, K.K., Meghvansi, M.K., Eds;). Springer, Cham 2022; Vol. 18, pp. 67-85
Reference n. 25. Cayuela, M.L.; Millner, P.D.; Meyer, S.L.F.; Roig, A. Potential of olive mill waste and compost as biobased pesticides against weeds, fungi, and nematodes. Sci. Total Environ. 2008, 399, 11-18.
Major points:
Taxonomic Nomenclature
The manuscript mentions several organisms (e.g., pepper, eggplant, nematodes) without providing their scientific names. According to scientific conventions, the Latin binomial should be included at the first mention of each species. For example:
Pepper — Capsicum annuum(or other species, if applicable)
Eggplant — Solanum melongena
Nematodes — The species/strain should be specified if possible (e.g., Meloidogyne incognitafor root-knot nematodes). If unknown, state "soil-dwelling nematodes (unspecified species)." Without this information, the reproducibility of the study is compromised.
The question has been addressed in the revised version
Inconsistent Abbreviations
The abbreviation "FWC1" is sometimes written as "FCW1" or "FWG1" (Table 1, line 166). This inconsistency must be corrected.
FCW1 appearing on Table 2 has been corrected, FWG1 was never written in the main text
Multiple abbreviations (e.g., HR, I-ERM, ROS, HA, FA, DH) are introduced but used only once, reducing readability. I would recommend to remove unnecessary abbreviations and use full terms, or keep only those repeated frequently (≥3–5 times)
Ok, it has been corrected
The abbreviation "PGP" (PGP traits, PGP effect) is not explained.
It was done in revised version
Experimental Design and Presentation
The experiment is extremely difficult to interpret due to the inconsistency of its implementation and presentation of the results. Thus, the authors use different doses of compost for eggplant (possibly 6 g kg-1), pepper (6 g kg-1, 10 g kg-1) and tomato (10 g kg-1, 30 g kg-1) and explain this by the fact that they considered these doses to be effective. However, it is unclear how the effective dose was determined.
A dose is effective when it works with nematodes (significant reduction of infection factors) and does not inhibit plant growth, ten different doses were previously screened for FWC1 – for the other formulations we have been doing hundreds of experiments by which we identified the effective doses
There is very little information about the experiment with eggplant plants. However, it is clear that preparations containing glomus fungi were not used in the experiments with eggplants.
Exactly
The experiment with tomato plants is more complete. It contains information about plants of different ages and also about the effect of adding FCW1 together with Ozor on nematode infestation. However, there is no data on the effect of Ozor on the tomato plant and pathogenic nematodes by itself.
We made experiments on the effect of Ozor on pepper plants; effects of Ozor on tomato plants has been shown on: Molinari, S.; Leonetti, P. Resistance to Plant Parasites in Tomato Is Induced by Soil Enrichment with Specific Bacterial and Fungal Rhizosphere Microbiome. Int. J. Mol. Sci. 2023, 24, 15416. doi: 10.3390/ijms242015416
Then we read about the experiment with pepper plants and see that a special experiment was conducted for pepper plants with Myco and Flortis preparations containing glomus fungi. Surprisingly, the authors did not include Ozor in this experiment, despite the fact that in the next experiment with pepper plants, mixtures of Ozor, Myco and Flortis preparations with the studied FCW1 were used .
As a result, it is difficult for the reader to interpret the data, because the work is very hard to assemble into a single picture and looks more like a patchwork. The problem is complicated by the fact that when compiling tables 1, 3, 4, some statistical methods were used (comparison only with the control), while in table 2, others were used (comparison of data from columns in pairs).
The submitted paper focuses on the effect of FWC1 – we wanted to observe if adding low amounts of AMF to those that are already contained in FWC1 supported FWC1 effects – we tested first Ozor on tomato because this formulation contains ONLY AMF – then we added low amounts of Myco and Flortis to FWC1 and tested the mix on pepper (we already had good results with the mix FWC1/Ozor). At this point, we wanted to know if this new compost as such was better than the formulations we have been using for long time in the past – this experiment was carried out with pepper. We think that the experimental design is as simple as described above. As far as the statistics, in Table 1, there are just one control and one treatment (mixes were tested AFTER experiments with the sole FWC1): in Table 2, we needed to know the significance of the difference also between the Myco/FWC1 and Flortis/FWC1 mixes. In Table 3, a PGP effect was observed ONLY by FWC1 treatment and effects on nematode infection were substantially the same for all the four formulations; we considered useless to make comparisons among formulations because it is evident there are not, however, if Rev. 4 thinks it is necessary we will perform a Duncan test also on these data; in Table 4, again, there are one control and one treatment.
In order to make it easier for the reader to read the article, the structure of the work would be changed. Thus, it is necessary to divide the work into parts. The first part should contain experiments with eggplant, tomato and pepper plants, which describe the effect of using FCW1 compared to control plants. The second part should contain data on the effect of preparations with glomus fungi on their own and in combination with FCW1. Here, you can also provide illustrations with microphotographs.
The third part is needed to combine the data from parts one and two using statistical methods. It would also be appropriate to provide data on gene expression here (Figure 4), and in conclusion, add the information described in Figure 3:
Effects of FCW1 alone (eggplant, tomato, pepper vs. controls).
Effects of Glomus fungi preparations (Myco, Flortis, Ozor ± FCW1), with microphotographs.
Integrated analysis combining Parts 1–2, including gene expression (Fig. 4) and the information described in Figure 3.
We appreciate the attempt of Rev. 4 to improve the structure of our paper. Unfortunately, Rev. 4 does not take into account that this study has been structured according to the results obtained on time, the schedule was:
1. Effect of different FWC1 dose on plants according to plant age (we had published papers that suggested this correlation) – it was a YES OR NO response
2. What happens if we mix effective doses of FWC1 with additional minimal amounts of AMF? Good or not?
3. is FWC1 a better choice as biofertilizer than the microbiome-generating formulations we have been using so far? And what about the performance on nematode infection?
4. What happens if we save the previously treated soil for the next crop? Will it still work?
Once we have shown that FWC1 is a good choice as biofertilizer and defense activator against nematodes, we provide data on the Mechanisms by Which FWC1 Acts as a Bio-Activator of Plant Defense Against RKNs showing that roots are colonized by AMF exactly as they are when only AMF are used. Afterwards, we show that this colonization induces some key genes associated with plant immune response.
That’s it. We have taken into account this schedule arranged experiment by experiment and we trust readers will not consider it as very complicated.
Minor points:
Misleading Terminology
The term «adult plants» for tomato plants weighing 5–7 g is misleading, as this stage typically represents the juvenile vegetative phase, not reproductive maturity. I recommend using «juvenile plants» or «vegetative-stage plants» instead. Similarly, «plantlets» could be replaced with «seedlings» if referring to very young plants (1.5–2 g) with cotyledons or early true leaves.
It has been done in the revised version
Formatting Issues in Tables
Table 2: The first column is too narrow, causing unit labels (e.g., "g⁻¹") to break across lines.
ok
Table 3: In the "+FCW1" column, some values repeat those in Table 2 (e.g. 84±21 (-42), 140±52 (-35), 586±209, 109±33 (-26)), while others do not (e.g. 30±10, 10.6±4.3 (40), 2.2±1.0 (51)). This inconsistency needs clarification.
The referee is absolutely right. There was in Tab. 3 a repetition of nematode infection data from Tab. 2 by mistake. In the revised version, the right data have been written on Tab. 3. Thanks for the report.
Round 2
Reviewer 1 Report
Comments and Suggestions for Authors
Authors have positively responded to my questions in the revision.
Author Response
Authors have positively responded to my questions in the revision.
thanks a lot
Reviewer 3 Report
Comments and Suggestions for Authors
It is still limited to that kind of material from that specific source. So, no further generalization can be foreseen.
Any research anticipated to be repeatable should be based upon strictly described input and output materials. Pre-mature, Halfway mature, and Fully mature composted material should be defined by its physical and chemical nature, not by general descriptive terms. Otherwise, it stays as some irrepeatable incidental trail.

Author Response
It is still limited to that kind of material from that specific source. So, no further generalization can be foreseen.
Any research anticipated to be repeatable should be based upon strictly described input and output materials. Pre-mature, Halfway mature, and Fully mature composted material should be defined by its physical and chemical nature, not by general descriptive terms. Otherwise, it stays as some irrepeatable incidental trail.
Tab. 5 describes the physical and chemical composition of the compost - no generalization has been put forward, we describe only FWC1
Reviewer 4 Report
Comments and Suggestions for Authors
I would like to thank the authors for the improving the manuscript, but some concerns remain to be addressed.
L204: correct "FGW1"
Table 4, 5: "FCW1" to "FWC1"
L645-647: please remove template text.
Author Response
I would like to thank the authors for the improving the manuscript, but some concerns remain to be addressed.
L204: correct "FGW1"
in the revised version at L 204 it is written FWC1
Table 4, 5: "FCW1" to "FWC1"
done, thanks
L645-647: please remove template text.
done
Round 3
Reviewer 3 Report
Comments and Suggestions for Authors
The universality in the composting process defines what goes into the substrate, and a close follow-up on the chemical and biological processes. Also, the determination of the nature and composition of the finally composted material.
This work is limited to one resource of undefined material and is not consistent.
Author Response
Rev. 3 Comments and Suggestions for Authors
The universality in the composting process defines what goes into the substrate, and a close follow-up on the chemical and biological processes. Also, the determination of the nature and composition of the finally composted material.
This work is limited to one resource of undefined material and is not consistent.
Authors: for food waste composts we cannot describe the exact composition of the compost - the repeatability is described in the way we are producing our compost. This is described in:
lines 432-438: For this work, the EC was loaded daily (from Monday to Friday) with 15–20 kg of biowaste and 2.2–5.5 kg of bulking agent (pruning of Arundo donax canes, shredded in pieces of 2-5 cm) which represented 15–20% wt of the food waste) for 60 days and completely emptied on the 91st day to ensure an average residence time for all the organic matter. At the beginning of the experiment, 25 kg of mature compost was uniformly distributed in-side the EC chamber as an inoculum of microorganisms. The biowaste used consisted of leftovers and kitchen scraps from the research center canteen. The FWC1 curing phase consisted of periodical overturn and watering of the heap for an additional 120 days. Finally, the mature compost was sieved using a Scheppach RS350 automatic rotary sieve fitted with a 10 mm mesh. The sieved compost was used for experiments and for physico-chemical characterization.
Composition in micro-elements, chemo-physical parameters, organic matter content and elemental analysis of FCWC1 are described in Table 5. A different compost which has the same characteristics should have the same effects on vegetable plant resistance to root-knot nematodes
